# Synergistic Effects of Zinc Oxide Nanoparticles and Moringa Leaf Extracts on Drought Tolerance and Productivity of *Cucurbita pepo* L. Under Saline Conditions

**DOI:** 10.3390/plants14040544

**Published:** 2025-02-10

**Authors:** Abdelsattar Abdelkhalik, Mohammed A. H. Gyushi, Saad M. Howladar, Abeer M. Kutby, Nouf A. Asiri, Areej A. Baeshen, Aziza M. Nahari, Hameed Alsamadany, Wael M. Semida

**Affiliations:** 1Horticulture Department, Faculty of Agriculture, Fayoum University, Fayoum 63514, Egypt; aga04@fayoum.edu.eg (A.A.); mah09@fayoum.edu.eg (M.A.H.G.); 2Department of Biological Sciences, College of Science, University of Jeddah, Jeddah 21959, Saudi Arabia; smhowladar@uj.edu.sa (S.M.H.); amkutby@uj.edu.sa (A.M.K.); naasiri@uj.edu.sa (N.A.A.); aabaeshen@uj.edu.sa (A.A.B.); amnahary@uj.edu.sa (A.M.N.); 3Department of Biological Sciences, Faculty of Science, King Abdulaziz University, Jeddah 21589, Saudi Arabia; halsamadani@kau.edu.sa

**Keywords:** nano fertilizers, biostimulants, antioxidants, osmoprotectants, *Cucurbita pepo*

## Abstract

This study investigated the combined effects of zinc oxide nanoparticles (Nano-Zn) and moringa leaf extract (MLE) on squash plants grown under water stress conditions in saline soil during 2021–2022. The research compared full irrigation (100% ETc) with water deficit conditions (60% ETc). While water deficit negatively impacted plant growth, yield, and various physiological parameters, the sequential application of Nano-Zn (at 50 or 100 mg L^−1^) with MLE (3%) significantly mitigated these adverse effects. The combined treatment proved more effective than individual applications, enhancing growth parameters, photosynthetic efficiency, and antioxidant systems. The treatment particularly improved stress tolerance by increasing protective compounds like soluble sugars and amino acids while reducing harmful H_2_O_2_ levels. The study concluded that sequential application of 100 mg L^−1^ Nano-Zn with MLE was optimal for enhancing squash performance under drought stress, with 50 mg L^−1^ Nano-Zn plus MLE as the second-best option.

## 1. Introduction

Among the Cucurbitaceous crops, squash (*Cucurbita pepo* L.) is a valuable fruit vegetable with high economic importance worldwide [1]. The squash plant has a shallow root system and is sensitive to soil water deficits, markedly decreasing fruit yield [2]. Therefore, squash plants need the water depletion level at the effective rooting depth to remain above 0.50 of the soil’s water-holding capacity [3]. Furthermore, squash is considered a moderately salt-tolerant crop and can suffer serious damage in saline environments, as salinity is a harsh factor that restricts the growth and yield of squash. [4].

Irrigation water is a crucial component of crop production since agriculture utilizes approximately 70% of freshwater resources [5]. However, in recent years, dry and semi-arid regions have experienced a severe freshwater shortage, endangering the production of crops, particularly those that are sensitive to water deficits [6]. In the coming years, it is expected that the need for freshwater will double with population growth, rising food demand, and climate change [7]. Therefore, a few years ago, the deficit irrigation technique was suggested, in which crops are irrigated below their irrigation water requirement to maximize water use efficiency (WUE) [8]. Nevertheless, it is challenging to apply DI without severe yield losses in arid regions that are susceptible to several environmental stresses such as salinity and drought [9]. Therefore, plant growth, productivity, and tolerance to environmental challenges are currently the focus of cutting-edge agricultural and plant-related technologies [10].

Drought and salinity are the most frequent abiotic stressors that severely reduce the productivity and development of plants. However, the concurrent occurrence of drought and salinity stress may increase the serious repercussions on plants, causing significant loss of crop yield [11,12]. Mostly, both drought and salinity stress have similar physiological effects on plants, causing osmotic and oxidative stress in tissues [13]. Soil water deficits first induce osmotic stress due to a drop in cell turgor pressure, which alters multiple cellular signaling pathways, resulting in cell dehydration, stomatal closure, and a reduction in CO_2_ influx into the leaf mesophyll cells [9]. Long-term drought exposure leads to a higher accumulation of free radicals (ROS; H_2_O_2_, OH^−^, and O2^•−^), which exacerbates oxidative damage to DNA, proteins, and lipid peroxidation [11]. Furthermore, these free radicals inactivate many important enzymes, reduce membrane integrity and selectivity, as well as trigger chlorophyll degradation and photoinhibition of the photosystem II (PSII) [9,14]. Soil water deficits lower nutrient uptake since nutrients are directly absorbed with the water by the roots, disturbing cellular ion homeostasis and several physio-biochemical functions [12,15]. The typical signs of drought stress in plants include leaf rolling, stunting, yellowing and scorching of leaves, and persistent wilting [16].

Plants have adaptive strategies to cope with stresses, including the accumulation of osmolytes involved in osmotic adjustment by maintaining cell turgor, facilitating water uptake, increasing relative water content, and even protecting macromolecules from ROS-induced damage [17,18]. Additionally, plants have antioxidative components, both enzymatic and non-enzymatic, that they use as a defense mechanism to protect the vital cell components against oxidative stress [19]. However, severe and prolonged stressors perturbed the natural equilibrium between the generation and scavenging of free radicals [15,20]. As a result, the exogenous application of sustainable and low-cost techniques such as nano fertilizers and/or biostimulants could lessen the deleterious consequences of abiotic stresses.

Engineered nanoparticles (NPs) are characterized by an innovative small size (1–100 nm) that gives them unique features such as a large surface area-to-volume ratio, greater adsorption efficiency, and increased reactivity [21,22,23]. Using nanoparticle-based fertilizers as a novel technique in the agriculture sector has recently gained great interest as an alternative to traditional fertilizers. Nano fertilizers have the potential to provide plants with nourishment, particularly micronutrients, and improve their ability to withstand abiotic stress [24,25]. This nanoscale fertilizer has higher reactivity, penetration, and translocation within plant tissues than its bulk-scale counterparts, thus increasing nutrient efficiency [21]. Furthermore, nano fertilizers are utilized to release nutrients gradually while lowering soil contamination [26].

Zinc (Zn) is a vital micronutrient that participates in several physiological activities required for plant growth and development [26]. Zn is involved in the catalytic activity of several enzymes like polymerases, RNA, dehydrogenases, aldolases, isomerases, and transphosphorylases [27]. It also contributes to the biosynthesis of enzymes, proteins, chlorophyll, and tryptophan, and plays a crucial role in maintaining membrane integrity, cell division, and metabolic turnover [25]. Recently, the application of Zn oxide (ZnO) in the form of nanoparticles (Nano-Zn) has been considered a practical way to enhance crop growth and production under various environmental stresses [24,28,29]. In drought-stressed cucumber, exogenously applied Nano-Zn enhanced the chlorophyll and carotenoid content, the PSII’s photochemical efficiency, and photosynthesis, along with increased biomass production, which is linked to an increase in osmolyte accumulation and the antioxidative defense system [25].

Biostimulants from plant extracts have garnered a lot of attention from the agro-industry and farmers because they are rich in physiologically active substances and can improve nutrient usage efficiency and raise crop production’s resistance to abiotic stress [30]. *Moringa oleifera* is one of the tropical regions’ most common tree species, which belongs to the Moringaceae family [28]. Moringa leaf extracts (MLE) are a naturally plentiful source of plant-enhancing substances like antioxidative compounds, soluble osmolytes, nutrients, and phytohormones, making it a potent plant biocatalyst [31]. Extracts of moringa’s leaves are rich in proline, sugars, amino acids, glutathione, salicylic acid, ascorbate, α-tocopherol, some vitamins, flavonoids, saponins, selenium, and carotene, and contain important quantities of auxin, cytokinin (zeatin), and gibberellins [28,31,32]. In previous research, foliar-applied MLE induced salt stress tolerance in *Silybum marianum*, thereby enhancing chlorophyll and carotenoid content, as well as promoting growth and fruit dry weight [33]. Exogenous MLE reduced oxidative stress markers while increasing the total soluble sugar content and glutathione contents of common bean grown under diverse salt and high-temperature stresses [34].

In the past couple of years, there has been an upsurge in interest in an emerging field of research that examines the synergetic effect of antioxidants/elicitors with biostimulants and how they affect plant physio-biochemical processes under abiotic stress [35,36]. In this study, squash plants grown in saline soil were subjected to drought stress and treated sequentially with Nano-Zn and MLE—methods that had never been used before under these conditions. Co-application of Nano-Zn and MLE may influence physio-biochemical changes that lessen the effects of salt and water stress on squash plants. Therefore, the present study aimed to explore the effect of Nano-Zn and/or MLE on the photosynthetic pigments and efficiency, tissue water content, membrane stability, osmoprotectants, non-enzymatic antioxidant contents, enzymatic antioxidant activities, growth, fruit yield, and water use efficiency of squash sown in salty soil (EC = 6.45 dS m^−1^) and exposed to different irrigation regimes.

## 2. Results

### 2.1. Effect of Nano-Zn and MLE on Growth and Productivity of Drought-Stressed Squash

Table 1 and Table 2 show that the growth and productivity of squash plants grown in salty soil were significantly influenced by irrigation regime, exogenous MLE, Nano-Zn, and their interactions. Reducing irrigation up to 60% of the ETc substantially reduced squash growth parameters (leaf number, leaf area, plant dry weight), fruit number per plant, fruit weight, and fruit yield, while not improving the WUE. MLE-treated plants had higher values of the above-mentioned parameters compared to non-MLE-treated plants. Regarding exogenous Nano-Zn, there were significant differences between the Nano-Zn concentrations; the highest values were observed in Nano-Zn100 whereas the lowest values corresponded to Nano-Zn0. As for the irrigation, MLE, and Nano-Zn interaction, the application of FI×MLE^+^×Nano-Zn100 treatment to squash plants resulted in the highest growth traits, fruit number, weight, and yield, while the application of FI×MLE^−^×Nano-Zn0 treatment resulted in the lowest values. However, Nano-Zn100 applied in sequence with MLE to drought-stressed squash plants counteracted the negative impact of water stress and increased the number of leaves by 97.5 and 100.1%, the area of the leaves by 51.7 and 52%, the plant’s dry weight by 85.1 and 94.6%, the number of fruits by 79.3 and 71.4%, the fruit yield by 75.9 and 84.1%, and WUE by 75.8 and 84.2% in 2021 and 2022, respectively, compared to non-MLE-Nano-Zn-treated squash plants under water deficit. Fruit weight was not significantly affected by the interaction of the three tested factors.

### 2.2. Effect of Nano-Zn and MLE on H_2_O_2_ Concentration, Tissue Water Status, and Membrane Integrity of Drought-Stressed Squash

Squash plants subjected to drought stress in salty soil notably increased the concentration of H_2_O_2_ in their tissues and reduced the MSI and RWC compared to those grown under FI (Table 3). Regarding the MLE effect, foliar-applied MLE reduced the H_2_O_2_ level while increasing the MSI and RWC in the leaves of squash plants (Table 3). For the Nano-Zn effect, foliage-spraying Nano-Zn_50_ or Nano-Zn_100_ significantly lowered the H_2_O_2_ level and increased the MSI and RWC compared to non-sprayed squash plants (Table 3). Under water deficit conditions, sequenced application of Nano-Zn with MLE relieved the deleterious effects on stressed squash plants. Contextually, sequenced-applied 50 or 100 mg L^−1^ Nano-Zn with MLE-mediated reductions in the H_2_O_2_ content (by 25.7 and 28.4% in the 2021 season and by 28.1 and 30.2% in the 2022 season) and elevation of the MSI (by 36.7 and 45.0% in the 2021 season and by 41.5 and 50.2% in the 2022 season) and RWC (by 25.6 and 30.3% in the 2021 season and by 29.5 and 34.4% in the 2022 season) of stressed plants, respectively, compared to the respective control (DI×MLE^−^×Nano-Zn_0_; Table 3).

### 2.3. Effect of Nano-Zn and MLE on Photosynthetic Pigment Content, and Photosynthetic Efficiency of Drought-Stressed Squash

The data in Table 4 and Table 5 show significant differences generated by the irrigation regime, MLE, Nano-Zn, and the interaction of the tested factors. The results indicate that water deficits induced a reduction in the photosynthetic pigment content, as well as the chlorophyll fluorescence parameters (*Fv*/*Fm* and PI). Higher values of chlorophyll a, b, carotenoids, *Fv*/*Fm*, and PI were observed in MLE-sprayed plants as compared to the non-MLE-sprayed plants. It was shown that plants sprayed with MLE had higher levels of chlorophyll a, b, carotenoids, Fv/Fm, and PI than plants not sprayed with MLE. As the Nano-Zn concentration increased, the photosynthetic pigments and chlorophyll fluorescence characteristics increased as well. According to data on the interaction effect, exogenously applied MLE and Nano-Zn have synergetic effects on attenuating the adverse effects that DI has on the photosynthetic pigments and the efficiency of PSII. Sequenced application of Nano-Zn_100_ with MLE increased chlorophyll a by 58.3 and 73.6%, chlorophyll b by 142.1 and 149.1%, carotenoids by 176.0 and 192.8%, Fv/Fm by 12.9 and 16.6%, and PI by 45.0 and 56.0% in the 2021 and 2022 seasons, respectively, in relation to the corresponding control (DI×MLE^−^×Nano-Zn_0_).

### 2.4. Effect of Nano-Zn and MLE on Osmoprotectants Accumulation of Drought-Stressed Squash

Irrigation level, MLE, Nano-Zn, and their interaction were mediated modulation in osmoprotectants accumulation in squash plants (Table 6). Water-deficit-stressed squash plants had higher total soluble sugars and free amino acids than those irrigated with FI. MLE-supplemented plants had notably raised accumulation of the abovementioned osmoprotectants more than the non-MLE-treated plants. Exogenous Nano-Zn at 50 or 100 mg L^−1^ significantly increased the osmoprotectant content compared to non-supplemented plants. The integrative effects of irrigation level, MLE, and different Nano-Zn concentrations on total soluble sugars and free amino acid contents in squash plants were significant. The maximum increases in the evaluated osmoprotectants were obtained under the FI×MLE^+^×Nano-Zn_50_ treatment, whereas the DI×MLE^−^×Nano-Zn_0_ treatment resulted in the lowest values. Sequenced application of Nano-Zn (50 or 100 mg L^−1^) with MLE to water-deficit-stressed plants produced significant increases in the total soluble sugars (by 33.7 and 54.2%) and free amino acids (by 13.6 and 19.8%) of the seasonal average, respectively, compared to the corresponding control.

### 2.5. Effect of Nano-Zn and MLE on Antioxidative Defense System of Drought-Stressed Squash

Drought stress substantially elevated the total phenolic content, glutathione, and ascorbic acid contents more than under optimum irrigation conditions (Figure 1). However, the non-enzymatic antioxidants (i.e., total phenolic content, glutathione, and ascorbic acid) in water deficit-stressed plants were further enhanced by exogenous Nano-Zn and MLE. Foliage application of 50 or 100 mg L^−1^ and MLE to stressed squash plants improved the total phenolic content by 13.7% or 20.9% (seasonal average), glutathione by 13.6% or 20.0% (seasonal average), and ascorbic acids by 18.8% or 22.4% (seasonal average), respectively, compared to stressed plants not supplemented with Nano-Zn or MLE (Figure 1).

The activities of CAT, SOD, APX, and GR were upregulated upon squash plants’ exposure to a 40% water deficit (Figure 2). Nonetheless, externally applied Nano-Zn and MLE in sequence further reinforced the non-enzymatic antioxidant activity under drought stress conditions. MLE and Nan-Zn (50 mg L^−1^) supplementation to drought-stressed squash plants raised the activity of CAT by 27.5% (seasonal average), SOD by 29.0% (seasonal average), APX by 28.7% (seasonal average), and GR by 16.7% (seasonal average) compared to the DI×MLE^−^×Nano-Zn_0_ treatment (Figure 3). However, these increases in the aforementioned enzymatic antioxidants were 37.9% (season average), 36.8% (seasonal average), 40.9% (seasonal average), and 33.1% (seasonal average), respectively, when MLE and Nano-Zn (100 mg L^−1^) were applied to stressed squash plants compared to water-stressed plants not treated with MLE or Nano-Zn (DI×MLE^−^×Nano-Zn_0_).

## 3. Discussion

As a novel approach to lessen the effects of drought stress on squash plants seeded in salty soil, ZnO nanoparticles (Nano-Zn) were foliar-applied sequentially with moringa leaf extract (MLE) in the current study. Drought and salinity stress mediated negative impacts on the growth and productivity of squash, as reported earlier by [37]. Also confirmed in this research, water stress diminished water status, membrane stability, chlorophyll biosynthesis, and PSII efficiency and subsequently hampered the growth and fruit yield of squash seeded in saline soil (6.45 dS m^−1^). Nonetheless, exogenous Nano-Zn in sequence with MLE was effective in alleviating the drought-stress-induced damage to squash plants.

In this study, drought stress at 40% severely decreased yield components and all growth parameters. As a result of metabolic abnormalities caused by water and salinity stresses, such as reduced meristematic activity and cell elongation due to the reduction in cyclin-dependent kinase enzyme activity, along with high respiration rates as a result of high energy demands, plants’ ability to grow and production are adversely affected [10,31]. However, exogenous Nano-Zn plus MLE applied sequentially counteracted the drought-mediated deleterious effects and stimulated the dry biomass production, leaf area, leaf number, and yield under normal and stress circumstances, and yielded the highest WUE (Table 2 and Table 3). Similar to our observations, [1,26] found that Nano-Zn or MLE, respectively, enhanced drought-stressed *Dracocephalum kotschyi* and squash development and yield. Squash growth promotion conferred by sequenced-applied Nano-Zn and MLE may be attributed to improvements in the RWC, MSI (Table 4), photosynthetic pigments, and photosynthetic capacity of PSII (Table 5). Zn regulates some growth-linked hormonal substances, which are important for cell extension, differentiation, and division. Furthermore, MLE supplemented abundant amounts of phytohormones, osmoprotectants, antioxidants, and nutrients (Table 8) to relieve the double stress (drought and salinity) while also promoting growth, biomass production, and fruit yield. The presence of phytohormones in MLE, including indoleacetic acids (IAA), gibberellins, and zein, has been linked to increasing cell division and expansion, which in turn stimulates the growth and productivity of MLE-treated plants [33]. Interestingly, in this study, the WUE increased up to 75.5–84.3% due to sequentially applied 50 or 100 Nano-Zn plus MLE to drought-stressed squash plants. Similarly, WUE has been improved under water deficit in response to the application of MLE [1] or Nano-Zn [24].

In the current research, increased endogenous production of H_2_O_2_ as an oxidative stress inducer was observed in drought-stressed squash plants as a negative mechanism that damages cellular membranes through lipid peroxidation and reduces the cellular membrane integrity (MSI; Table 3). These unfavorable outcomes were positively modulated by leafy-applied Nano-Zn sequentially with MLE that may be enhanced pathways to reduce the H_2_O_2_ level in squash plants. Higher antioxidative molecules (enzymatic and non-enzymatic antioxidants, and proline) by exogenous Nano-Zn and MLE are associated with reduced ROS and malondialdehyde acetate content, as well as cell membrane permeability of linseed under Cd toxicity [28]. Therefore, stressed squash plants treated with Nano-Zn plus MLE restored the cellular membrane integrity, increasing the MSI and suggesting a Nano-Zn plus MLE-mediated drought stress tolerance.

Soil water deficit leads to IAA-mediated stomatal closure and decreases water uptake, both of which have an impact on several metabolic pathways, including the reduction in water stored in leaves [9], ultimately leading to a decline in the leaf RWC (Table 4). However, foliage spraying Nano-Zn plus MLE increased the RWC, a metabolically accessible water vital for preserving tissue water status and physiological activity [38]. The mechanism underlying Nano-Zn improvement of RWC in drought stress conditions may be connected to Zn’s effect on hormone induction that controls root development, water absorption, and transport capacity for improved adaptation to water deficits [39,40]. Also, previous research demonstrated that the application of MLE elevated osmoprotectant accumulation and RWC in *Cucurbita pepo* plants under water deficit [1]. Maintaining high tissue water contents has been observed by sequenced-applied MLE and GSH due to the accumulation of Ca^2+^, K^+^, soluble sugars, and proline [35].

Water stress induces chloroplast structure damage and instability of pigment-protein complex, and perturbs the chlorophyllase activity [27], in turn resulting in a decline in the leaf chlorophyll content and photoinhibition of the photochemical efficacy of the PSII, as evidenced in this study (Table 5). According to our results, exogenous Nano-Zn and MLE induced drought stress tolerance in squash plants, given that Nano-Zn and MLE mediated restoration in the chlorophylls and carotenoids concurrently with enhanced chlorophyll fluorescence apparatus (*Fv*/*Fm* and PI). Our findings are in harmony with those reported in eggplant [24] and bean [34] plants under environmental stresses. These beneficial changes in the pigments involved in photosynthetic processes and efficacy of the PSII of drought-stressed squash may be ascribed to enhanced leaf membrane stability and water content for maintaining chloroplast ultrastructure, as well as chlorophyll biosynthesis. In support of our findings, Nano-Zn increased the levels of *Fv*/*Fm*, PI [24], and chlorophyll [26] by preserving sulfhydryl (-SH) bonds in the cell membrane, balancing the concentration, and supplying other elements (N and Mg) necessary for chlorophyll production [41]. In addition, MLE is an excellent source of essential nutrients and amino acids (Table 8) that are necessary for the manufacture of chlorophyll. As a result, its presence boosts the capacity for photosynthetic activity in drought-stressed squash plants growing in salty soil.

Our results demonstrated that the co-occurrence of drought and salinity stress stimulated the pilling up of osmolytes (i.e., soluble sugars and amino acids; Table 6). However, the sequenced application of Nano-Zn with MLE further elevated the osmolyte contents in stressed squash, indicating their ability to improve the osmolyte accumulation. These compatible solutes play a key role in salt and drought stress tolerance through osmotic adjustment to preserve cell turgor and boost the RWC, as well as preventing oxidative damage to stabilize thylakoid membranes, reduce photoinhibition, and improve photosynthesis [6,42]. The present study showed that the increase in osmolytes by sequentially applied Nano-Zn with MLE correlated positively with enhanced squash growth and yield, which may be attributed to more photoassimilates, nutrients, and phytohormones. Our findings are consistent with those of [28], who stated that Nano-Zn and MLE had a synergistic effect on Cd stress tolerance by increasing proline, soluble sugars, and soluble protein accumulation. Moreover, MLE containing osmolytes (proline, amino acids, soluble sugars, and K^+^; Table 8) may direct osmotic adjustment as an efficient way to overcome water stress.

Under stressful conditions, plants activate their defense system, which contains enzymatic and non-enzymatic antioxidants, such as CAT, SOD, APX, GR, total phenolic content, GSH, and AsA [15], as evidenced in the present study (Figure 2). These antioxidative compounds scavenge ROS and restore cell redox homeostasis [18]; however, under severe stress conditions, the balance between production and scavenging ROS could be disturbed [43]. Nonetheless, applying Nano-Zn in sequence with MLE to water-stressed squash plants sown in salty soil boosted the antioxidant machinery, alleviating the stress effects. Clear increases in the activities of CAT, SOD, APX, and GR, along with total phenolic, GSH, and AsA contents were observed in Nano-Zn and MLE-treated plants under drought stress (Figure 2 and Figure 3). Our results are consistent with those for cucumber [25] and maize [44], which stated that Nano-Zn application mediated drought stress tolerance by elevating the antioxidant defense system followed by a discernible drop in H_2_O_2_. Zn is a cofactor of numerous enzymes [45]; thus, the use of Nano-Zn increased the activity of CAT, SOD, POD, GR, and APX while raising the AsA, GSH, and total phenol levels in cucumber under water deficit conditions [25]. A rich source of antioxidants, *Moringa oleifera* extract contains phenols, AsA, GSH, proline, salicylic acid, flavonoids, and α-tocopherol, which *Lepidium sativum* plants can absorb and use to enhance their own antioxidant system [46]. These enhanced antioxidant responses to Nano-Zn and MLE treatment on the leaves helped the plants scavenge the extra ROS produced in response to stressful signals.

Our results demonstrated that exogenously applied Nano-Zn in sequence with MLE was an innovative way to ameliorate drought-induced damage to squash plants sown in salty soil while also improving squash performance. Co-applied Nano-Zn and MLE showed higher efficacy than when individually applied.

## 4. Materials and Methods

### 4.1. Experimental Conditions, Plant Details, and Irrigation Water Requirements

Two-season open field experiments were performed during the spring season of 2021 and 2022 at the experimental station of the faculty of agriculture (29°17′36.4″ N 30°55′01.6″ E), Fayoum University, Fayoum, Egypt. The soil physicochemical analysis in Table 7 was carried out following the procedures of [47,48]. The experimental soil is saline (ECe = 6.45) and slightly alkaline (pH = 7.68). The climate conditions in the area are arid based on the aridity index [49].

Squash (*Cucurbita pepo*) seeds of the hybrid Yara were seeded on 15 March 2021 and 20 March 2022 at 0.5 m in beds of 1 m width. Squash plants were irrigated by a drip irrigation system with on lateral line of 16 mm per bed, and a dripper of 4 L per hour spaced 0.50 m apart at the same distance between plants. Plants were fertilized with the recommended local doses of mineral fertilizers following the recommendations of [1] at rates of 150 kg N ha^−1^, 60 P_2_O_5_ ha^−1^, and 120 K_2_O ha^−1^ through drip irrigation. The recommended quantities of N, P, and K were added at ratios of 5:1:3 during vegetative growth, 5.2.4 during the flowering stage, and 5:3:5 during the harvesting stage. Agronomic practices, as well as pest and disease control, were performed following the bulletin of squash production issued by the Egyptian Agricultural Research Center.

The crop irrigation needs (ETc) were computed based on the reference evapotranspiration obtained from Class A pan (E_pan_) placed in the experimental station, the pan coefficient (K_pan_), and the crop coefficient (K_c_), with the following equation [50]:ETc = E_pan_ × K_pan_ × K_c_

The squash crop coefficient (K_c_), according to [50] was 0.6, 0.95, and 0.75 during the initial, mid, and end seasons, respectively. Lengths of squash development stages were adapted in the local area to 21, 35, 25, and 15 days for initial, crop development, mid-season, and end-season stages, respectively.

The irrigation water requirements for each irrigation strategy were determined as follows:$$\mathrm{IWA}=\frac{A\times \mathrm{ETc}\times \mathrm{Ii}\times \mathrm{Kr}}{\mathrm{Ea}\times 1000\times (1-\mathrm{LR})}$$
where IWA is the irrigation water applied (m^3^), A is the plot area (m^2^), ETc is the crop water requirements (mm day^−1^), Ii is the irrigation intervals (day), Kr is the covering factor, Ea is the application efficiency (%), and LR is the leaching requirement.

At intervals of two days, squash plants were irrigated with varying amounts of irrigation water.

### 4.2. Treatments, and Experimental Design

Treatments of the experiments were arranged in a split–split plot design in RCBD with three factors: irrigation regimes (main plot), moringa leaf extract (subplot), and zinc oxide nanoparticles (sub-subplot). Two irrigation regimes included full irrigation (FI; 100% of the crop evapotranspiration = ETc) and deficit irrigation (DI; 60% of the ETc). The moringa leaf extract (MLE) was applied at 3% (MLE^+^) or not applied (MLE^−^). Three zinc oxide nanoparticle (Nano-Zn) concentrations were as follows: not applied (Nano-Zn_0_), 50 mg L^−1^ (Nano-Zn_50_), and 100 mg L^−1^ (Nano-Zn_100_). Therefore, twelve treatments were designed and triplicated, for a total of 36 plots, each consisting of two beds (8 m length and 1 m width) of approximately 16 m^2^. All experimental units were surrounded by a 1 m non-irrigated area. For good penetration into squash leaves, Tween 20 was added to the spraying solution upon application. The MLE and Nano-Zn were sequentially applied as foliar sprays as follows: Nano-Zn was applied at 15 and 25 days after sowing, while MLE was applied at 20 and 30 days after sowing. From sowing until a week after full germination, all treatments were watered with full irrigation requirements; after that, the irrigation treatments were established. Fully matured fresh leaves of *Moringa oleifera* trees were harvested and air dried, then grained and extracted [51]. Ethyl alcohol was mixed with the leaf powder and kept for 4 h in a rotary shaker; thereafter, the mixture was filtered twice by filter paper (Whatman No. 1) for purification. After that, the mixture was exposed to a rotary evaporator to evaporate all alcohol. Then, the centrifugation process was conducted (8000× *g* for 15 min) and the supernatant was diluted 30 times before spraying onto squash plants. Comprehensive chemical characterization of the extract was conducted, with a detailed compositional analysis presented in Table 8.

Zinc oxide nanoparticles (Sigma-Aldrich^®^, St. Louis, MO, USA) were purchased. To prepare Nano-Zn levels of Nano-Zn_50_ and Nano-Zn_100_, 50 and 100 mg L^−1^, respectively, were dissolved in double-distilled water. The suspensions were then subjected to ultra-sonication for 30 min to improve the dispersion of the nanoparticles. The characterization of Nan-Zn was performed by TEM (Figure 3).

### 4.3. Sampling and Measurements

Growth parameters were determined with three plants randomly taken from each experimental plot 60 days after sowing. The number of leaves per plant was recorded and the leaf area per plant was determined as reported by [6]. Ten disks of 1 cm^2^ were taken from the leaves, after cleaning them, and dried at 70 °C for 24 h. The plant leaf area was estimated using the following formula:$$\mathrm{Leaf}\;\mathrm{area}\;{\mathrm{plant}}^{-1}=\left[\frac{LDW}{DDW}\right]\times DA$$
where LDW refers to leaf dry weight (g) per plant, DDW is the leaf disk dry weight, and DA is the disk’s area.

The plants were dried at 70 °C in a forced oven until reaching a constant weight, then weighed dry.

Six plants per plot were determined to obtain the yield-related parameters. Harvests were undertaken from 4 May 2021 to 19 June 2021, and again from 8 May to 23 June 2022. The number of fruits per plant, average fruit weight, and total fruit weight were recorded each harvest. The cumulative fruit yield per ha was calculated at the end of the harvest. The water use efficiency (WUE) for each treatment was determined using the following formula [52]:$$\mathrm{WUE}\;(\mathrm{kg}\;m^{-3})=\frac{Fruit\;yield\;(kg{ha}^{-2})}{Irrigation\;water\;applied\;(m^{3}{ha}^{-1})}$$

### 4.4. Photosynthetic Pigments and Chlorophyll Fluorescence

Chlorophyll *a*, *b*, and carotenoid contents were determined using the dimethyl-formamide (DMF) method [53,54]. Three leaf discs (5 mm diameter) were taken from the fifth leaf of each plant. Leaf discs were stored in separate Eppendorf tubes with 1 mL of DMF and kept in the dark at 4 °C for >48 h. Chlorophyll *a* and *b* contents were measured by the absorption at wavelengths 647 nm and 664 nm using a spectrophotometer (UV-Visible Spectroscopy System, Hewlett Packard 95–98). The concentration of chlorophyll *a* and *b* was calculated according to the following formulas [54]:Chlorophyll *a* (Chl *a*) = 11.65 A664 − 2.69 A647Chlorophyll *b* (Chl *b*) = 20.81 A647 − 4.53 A664

Total carotenoid content was measured by the absorption at wavelength 480 nm via a spectrophotometer (UV-Visible Spectroscopy System, Hewlett Packard 95–98) and was calculated according to the following formula:Total carotenoids = [1000A480 − 0.89 (Chl *a*) − 52.02 (Chl *b*)]/245

The chlorophyll fluorescence measurements were performed on one leaf from each plant at solar noon using a handheld fluorometer (Handy PEA, Hansatech Instruments Ltd., Kings Lynn, UK). The plant’s leaves were dark-adapted for 30 min before measurements were taken with the fluorometer on intact leaves. According to [14], the maximum quantum yield of PSII (Leaf Fv/Fm) was calculated as the ratio of variable fluorescence (Fv) to the maximum chlorophyll fluorescence (Fm), and Fv = (Fm − F0). The calculation of the photosynthetic performance index (PI) was performed following the guidelines provided by [55].

### 4.5. Relative Water Content and Membrane Stability Index

The relative water content (RWC) of squash leaves was determined in 10 disks (2 cm^2^) prepared from fully developed leaves as described by [5] and the RWC was calculated as follows:$$\mathrm{RWC}\;(\%)=\left[\frac{DFW-DDW}{DTW-DDW}\right]\times 100$$
where the disks’ fresh weight taken from the leaves is DFW, the disks’ weight after soaking in water for 6 h is DTW, and the disks’ weight after drying in a forced oven at 70 °C for 24 h is DDW.

The determination procedures of membrane stability index (MSI) and the formula (MSI (%) = 1 − (EC_a_/EC_b_) × 100) of [56] were used. A squash leaf sample (0.2 g) was inserted into a test tube with 10 mL of double distilled water, and then the samples were heated in a water bath for 30 min at 40 °C. After that, EC_a_ was measured. Thereafter, the samples were again placed in the water bath for 10 min at 100 °C and the solution’s electrical conductivity was measured (EC_b_).

### 4.6. Osmoprotectants

The procedure outlined in [57] was used for squash leaf extraction (96% *v*/*v* ethanol) and for quantifying the content of soluble sugars (mg per g of leaf dry weight). After cooling the samples, the absorbance was recorded at 625 nm with UV-Vis spectrophotometer T80.

The amount of total free amino acids in squash leaves was assessed [58]. For extraction, 0.2 g of dried squash leaf tissues was placed in 10 mL of 80% ethanol and left for 48 h, then filtered. The solution was heated for 10 min in a water bath; after that, the samples were cooled, and then the absorbance was measured with UV-Vis spectrophotometer T80 at a wavelength of 570 nm.

The total phenolic content was quantified based on the Folin–Ciocalteu technique [59]: 1 mL of the phenolic extract was mixed with 1 mL of 10% Folin–Ciocalteu phenol reagent and 1 mL of 20% anhydrous sodium carbonate. After homogenizing the mixture with a shaker, the reaction was given 30 min to stabilize. The absorbance was taken spectrophotometrically at 750 nm. The calculation procedures were performed based on comparing the absorbance with a gallic acid standard curve.

### 4.7. Oxidative Stress Indicator (H_2_O_2_)

The hydrogen peroxide (H_2_O_2_) level (μmol g^−1^ FW) was estimated as described by [60]. In brief, 250 mg of fresh squash leaf was mashed in 5 mL of trichloroacetic acid, 5% (TCA), and the resultant was centrifuged (12,000× *g*, 15 min, 4 °C). The harvested supernatants were mixed well with a reaction medium composed of 1 M KI + 10 mM K-phosphate buffer (pH 7.0). The optical density was taken at 390 nm against H_2_O_2_ as a standard.

### 4.8. Enzymatic and Non-Enzymatic Antioxidants

The methods described by [61] were used to determine the glutathione (GSH) content. The optical density was measured at 412 nm, and the levels of GSH were calculated using a standard curve. The Folin phenol reagent was used to evaluate the ascorbic acid (AsA) [62]. Oxalis acid was used to grind 500 mg of dry squash leaf before centrifuging it at 15,000× *g* for 5 min at 4 °C. Following vigorous shaking, the diluted Folin reagent (0.2 mL) was added to the diluted supernatant (0.2–0.5 mL dilution to 2 mL), and the absorbance at 760 nm was measured.

For the extraction of antioxidant enzymes [63], 0.5 g of fresh leaf sample was homogenized in ice-cold 0.1 M phosphate buffer (pH = 7.5) containing 0.5 mM EDTA with pre-chilled pestle and mortar. After centrifuging the homogenate at 4 °C for 15 min at 15,000× *g*, the supernatant was collected to test the enzyme activity. The activity of superoxide dismutase (SOD; U mg^−1^ protein) was quantified by monitoring the enzyme’s reduction in the absorbance of the superoxide-nitro blue tetrazolium complex [64]. The catalase (CAT; U mg^−1^ protein) activity was assessed by observing the reduction in absorbance at 240 nm mediated by H_2_O_2_ decomposition [65]. The glutathione reductase (GR; U mg^−1^ protein) activity was quantified [66] by noting the increase in absorbance (412 nm for 5 min) in the reaction mixture containing 0.1 mL enzyme extract, 1 mL of 0.2 M K-phosphate buffer (pH = 7.5) containing 0.1 mM EDTA, 0.5 mL of 3 mM DTNB in 0.01 M K-phosphate buffer (pH = 7.5), 0.1 mL of 2mM NADPH, and adding 0.1 mL of 2 mM GSSG. The ascorbate peroxidase (APX; U mg^−1^ protein) activity was determined by recording the rate of AsA oxidation at 290 nm.

### 4.9. Statistical Analysis

GenStat statistical tool (version 12; VSN International Ltd., Oxford, UK) was used to conduct the analysis for the two seasons based on RCBD. Means for all variables were separated using Fisher’s least-significant difference test at *p* ≤ 0.05.

## 5. Conclusions

Drought stress noticeably increased the H_2_O_2_ level along with reduced growth, fruit yield, and physio-biochemical attributes of squash sown in saline soil. However, sequentially applied zinc oxide nanoparticles (Nano-Zn) with moringa leaf extracts (MLE) enhanced the ability of squash plants to withstand drought stress. Dry biomass production, leaf area, fruit yield, and water use efficiency of squash plants grown under water deficit were improved by foliar application of Nano-Zn and MLE. Synergetic application of Nano-Zn and MLE increased membrane stability and relative water content, and improved chlorophyll and carotenoid production and photosynthetic efficiency of the PSII (Fv/Fm and PI). The osmolyte accumulation (soluble sugars, proline, and amino acids) and the activity of antioxidants were elevated by foliar-applied Nano-Zn in sequence with MLE to ameliorate the osmotic stress while reducing the level of H_2_O_2_ to mitigate the oxidative stress.

## Figures and Tables

**Figure 1 plants-14-00544-f001:**
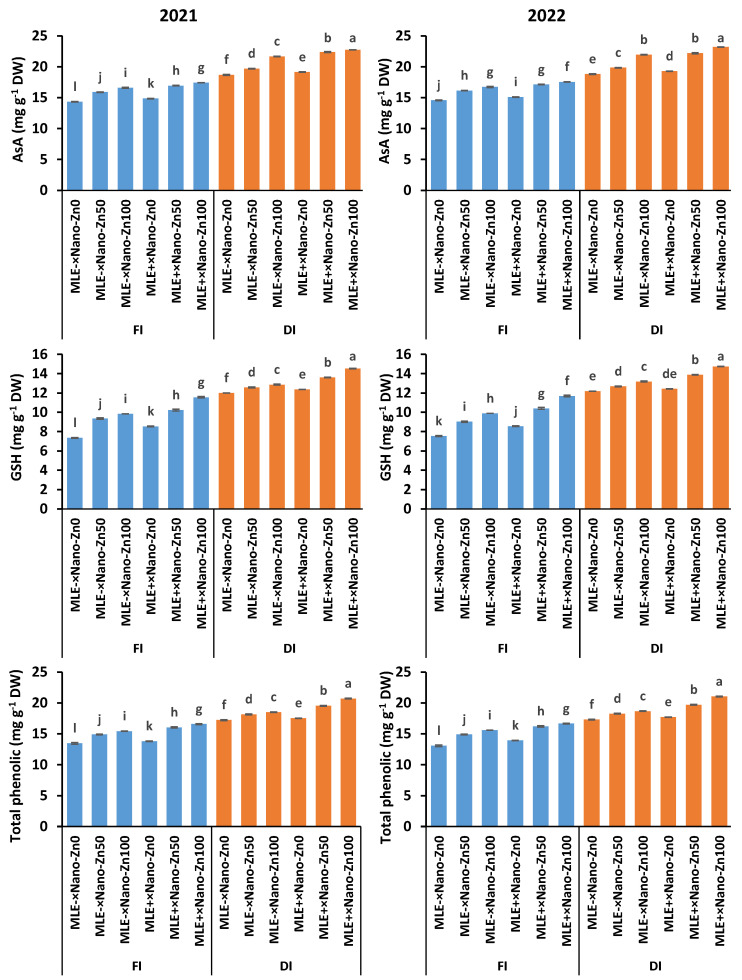
Non-enzymatic antioxidant (total phenolic content; glutathione, GSH; ascorbic acid, AsA) responses to exogenously applied ZnO nanoparticles (Nano-Zn) and moringa leaf extract (MLE) under normal (FI) and drought (DI) conditions during the 2021 and 2022 seasons, columns (±SE bar) with different letters are significantly different.

**Figure 2 plants-14-00544-f002:**
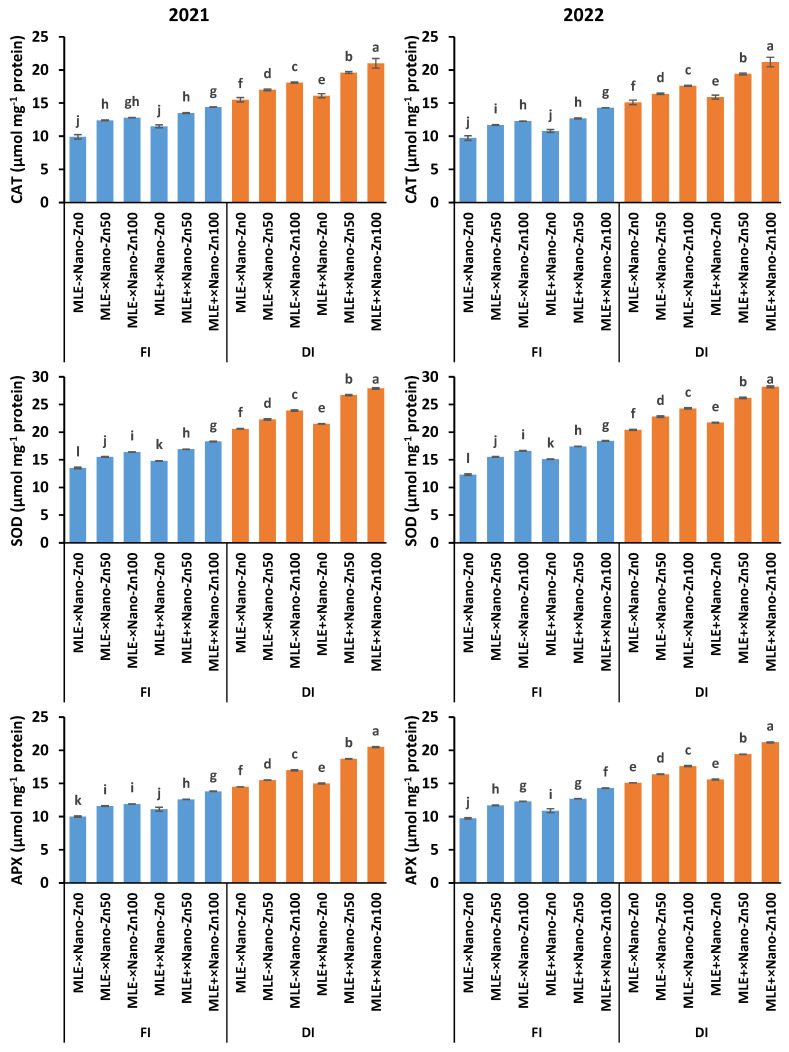
Enzymatic antioxidant (catalase, CAT; superoxide dismutase, SOD; ascorbate peroxidase, APX; glutathione reductase, GR) responses to exogenously applied ZnO nanoparticles (Nano-Zn) and moringa leaf extract (MLE) under normal (FI) and drought (DI) conditions during the 2021 and 2022 seasons, columns (±SE bar) with different letters are significantly different.

**Figure 3 plants-14-00544-f003:**
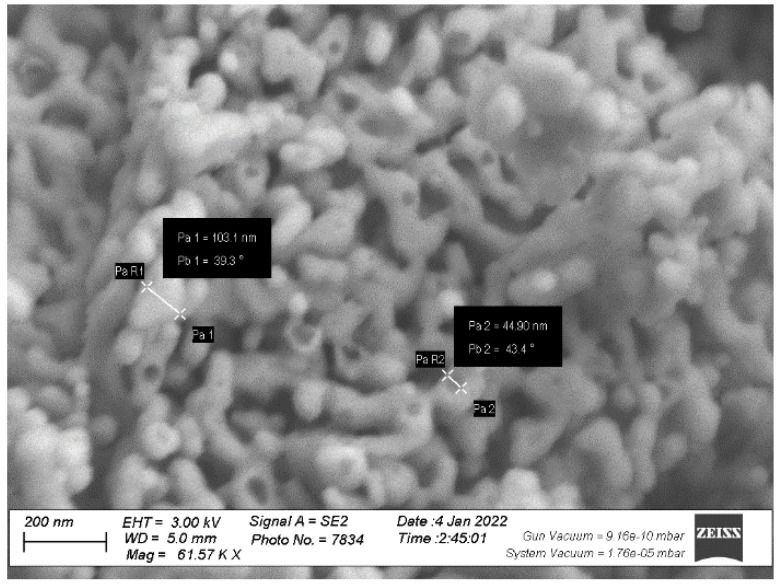
Transmission electron microscopy image of Zn oxide nanoparticles.

**Table 1 plants-14-00544-t001:** Squash growth attribute responses to exogenously applied ZnO nanoparticles (Nano-Zn) and moringa leaf extract (MLE) under normal (FI) and drought (DI) conditions during the 2021 and 2022 seasons.

Treatments	Leaf Number Plant^−1^		Leaf Area DM (decimeter)		Plant Dry Weight (g plant^−1^)	
	2021	2022	2021	2022	2021	2022
**Irrigation**						
FI	20.1 ± 1.11 a ^#^	24.0 ± 1.13 a	416.7 ± 5.94 a	401.6 ± 5.94 a	232.3 ± 5.46 a	218.1 ± 5.57 a
DI	19.5 ± 1.02 b	16.7 ± 0.94 b	368.4 ± 11.3 b	356.1 ± 11.0 b	174.1 ± 8.54 b	162.9 ± 8.65 b
**MLE**						
MLE^−^	20.4 ± 1.08 b	18.1 ± 1.08 b	374.6 ± 11.7 b	361.3 ± 11.2 b	189.0 ± 9.87 b	176.1 ± 9.48 b
MLE^+^	25.1 ± 1.32 a	22.6 ± 1.40 a	410.1 ± 7.53 a	396.5 ± 7.40 a	217.4 ± 9.02 a	204.9 ± 9.01 a
**Nano-Zn**						
Nano-Zn0	17.8 ± 1.09 c	15.6 ± 1.09 c	351.3 ± 13.5 c	338.3 ± 12.8 c	168.8 ± 11.4 c	156.0 ± 10.9 c
Nano-Zn25	24.0 ± 1.20 b	21.5 ± 1.32 b	405.4 ± 7.73 b	391.4 ± 7.50 b	214.1 ± 9.49 b	200.6 ± 9.25 b
Nano-Zn50	26.5 ± 1.40 a	24.0 ± 1.52 a	421.1 ± 6.93 a	407.1 ± 6.70 a	226.7 ± 8.98 a	214.9 ± 8.84 a
**Irrigation×MLE×Nano-Zn**						
FI×MLE^−^×Nano-Zn0	19.3 ± 0.33 fg	17.0 ± 0.58 f	376.6 ± 1.42 i	361.6 ± 1.40 h	194.1 ± ±1.87 h	178.7 ± 4.16 g
FI×MLE^−^×Nano-Zn50	24.7 ± 0.33 d	22.7 ± 0.33 d	413.9 ± 0.69 e	398.9 ± 1.69 d	231.4 ± 0.45 d	217.4 ± 0.45 d
FI×MLE^−^×Nano-Zn100	26.7 ± 0.33 c	24.7 ± 0.33 c	427.6 ± 1.15 c	412.6 ± 1.15 c	241.4 ± 1.16 c	227.4 ± 1.16 c
FI×MLE^+^×Nano-Zn0	22.7 ± 0.33 e	20.7 ± 0.33 e	396.4 ± 2.24 g	381.4 ± 2.22 f	215.4 ± 1.37 f	201.4 ± 1.37 e
FI×MLE^+^×Nano-Zn50	29.7 ± 0.33 b	27.7 ± 0.33 b	436.5 ± 0.47 b	421.5 ± 1.57 b	249.9 ± 1.55 b	235.9 ± 1.55 b
FI×MLE^+^×Nano-Zn100	33.3 ± 0.33 a	31.3 ± 0.33 a	448.9 ± 1.45 a	433.9 ± 1.45 a	261.9 ± 1.19 a	247.9 ± 1.19 a
DI×MLE^−^×Nano-Zn0	13.0 ± 0.58 i	11.3 ± 0.33 i	278.3 ± 2.48 l	269.3 ± 2.38 k	120.1 ± 0.96 l	111.1 ± 1.83 k
DI×MLE^−^×Nano-Zn50	18.7 ± 0.33 g	15.7 ± 0.33 g	365.7 ± 1.36 j	352.7 ± 2.36 i	165.7 ± 1.94 j	153.7 ± 0.93 i
DI×MLE^−^×Nano-Zn100	20.3 ± 0.33 f	17.3 ± 0.33 f	385.4 ± 2.01 h	372.4 ± 2.51 g	181.2 ± 0.44 i	168.2 ± 0.95 h
DI×MLE^+^×Nano-Zn0	16.3 ± 0.33 h	13.3 ± 0.33 h	353.7 ± 1.93 k	340.7 ± 2.93 j	145.7 ± 1.22 k	132.7 ± 0.56 j
DI×MLE^+^×Nano-Zn50	23.0 ± 0.58 e	20.0 ± 0.33 e	405.2 ± 2.34 f	392.2 ± 1.35 e	209.5 ± 2.18 g	195.5 ± 2.18 f
DI×MLE^+^×Nano-Zn100	25.7 ± 0.33 cd	22.7 ± 0.33 d	422.3 ± 1.07 d	409.3 ± 1.77 c	222.2 ± 0.85 e	216.2 ± 1.75 d

^#^ Values are means of (n = 5) ± standard error. Mean values in each column followed by a different lowercase letter are significantly different by Duncan’s multiple range test at *p* ≤ 0.05.

**Table 2 plants-14-00544-t002:** Yield-linked parameters and water use efficiency (WUE) responses to exogenously applied ZnO nanoparticles (Nano-Zn) and moringa leaf extract (MLE) under normal (FI) and drought (DI) conditions during the 2021 and 2022 seasons.

Source of Variation	Fruits Number Plant^−1^		Fruit Weight (g)		Yield (ton ha^−1^)		WUE (Kg m^−3^)	
	2021	2022	2021	2022	2021	2022	2021	2022
**Irrigation**								
FI	10.67 ± 0.36 a ^#^	10.47 ± 0.37 a	79.8 ± 0.58 a	78.8 ± 0.53 a	16.9 ± 0.49 a	16.60 ± 0.68 a	4.12 ± 0.12 b	4.52 ± 0.19 b
DI	8.55 ± 0.36 b	8.31 ± 0.33 b	78.5 ± 0.55 b	77.7 ± 0.50 b	13.16 ± 0.55 b	13.07 ± 0.59 b	4.97 ± 0.21 a	5.45 ± 0.25 a
**MLE**								
MLE^−^	8.81 ± 0.40 b	8.64 ± 0.39 b	78.4 ± 0.54 b	77.6 ± 0.46 b	14.02 ± 0.68 b	13.49 ± 0.65 b	4.23 ± 0.19 b	4.52 ± 0.19 b
MLE^+^	10.41 ± 0.40 a	10.13 ± 0.41 a	79.9 ± 0.58 a	78.9 ± 0.55 a	16.0 ± 0.60 a	16.18 ± 0.74 a	4.86 ± 0.17 a	5.45 ± 0.24 a
**Nano-Zn**								
Nano-Zn0	7.99 ± 0.41 c	7.82 ± 0.37 c	76.4 ± 0.47 c	75.8 ± 0.41 c	12.54 ± 0.75 c	11.90 ± 0.61 c	3.76 ± 0.17 c	3.98 ± 0.16 c
Nano-Zn25	9.91 ± 0.34 b	9.65 ± 0.35 b	80.0 ± 0.27 b	79.0 ± 0.35 b	15.9 ± 0.56 b	15.41 ± 0.57 b	4.82 ± 0.18 b	5.21 ± 0.23 b
Nano-Zn50	10.93 ± 0.49 a	10.69 ± 0.49 a	81.1 ± 0.48 a	80.0 ± 0.38 a	16.7 ± 0.67 a	17.19 ± 0.86 a	5.05 ± 0.18 a	5.76 ± 0.12 a
**Irrigation×MLE×Nano-Zn**								
FI×MLE^−^×Nano-Zn0	8.67 ± 0.04 f	8.33 ± 0.04 e	76.1 ± 0.44 a	75.44 ± 0.59 a	13.72 ± 0.11 f	12.71 ± 0.06 h	3.36 ± 0.03 i	3.46 ± 0.02 j
FI×MLE^−^×Nano-Zn50	10.21 ± 0.09 cd	10.11 ± 0.02 c	79.5 ± 0.81 a	78.16 ± 0.97 a	16.5 ± 0.27 ac	15.87 ± 0.13 d	4.04 ± 0.07 gh	4.33 ± 0.04 g
FI×MLE^−^×Nano-Zn100	11.06 ± 0.08 b	10.86 ± 0.08 b	80.5 ± 0.59 a	79.46 ± 0.59 a	17.39 ± 0.36 c	17.25 ± 0.15 c	4.25 ± 0.09 fg	4.70 ± 0.04 f
FI×MLE^+^×Nano-Zn0	9.44 ± 0.10 e	9.24 ± 0.10 d	78.7 ± 0.23 a	77.74 ± 0.23 a	15.38 ± 0.24 d	14.37 ± 0.12 f	3.76 ± 0.06 h	3.92 ± 0.03 h
FI×MLE^+^×Nano-Zn50	11.30 ± 0.15 b	11.13 ± 0.13 b	81.0 ± 0.37 a	80.33 ± 0.58 a	18.43 ± 0.25 b	17.81 ± 0.27 b	4.51 ± 0.06 ef	4.85 ± 0.07 f
FI×MLE^+^×Nano-Zn100	13.33 ± 0.17 a	13.13 ± 0.17 a	83.3 ± 1.12 a	81.60 ± 0.80 a	19.75 ± 0.44 a	21.56 ± 0.19 a	4.83 ± 0.11 de	5.88 ± 0.05 c
DI×MLE^−^×Nano-Zn_0_	5.83 ± 0.14 h	5.90 ± 0.13 g	74.9 ± 0.42 a	74.92 ± 0.40 a	8.74 ± 0.29 h	8.84 ± 0.21 j	3.30 ± 0.11 i	3.68 ± 0.09 i
DI×MLE^−^×Nano-Zn50	8.22 ± 0.04 g	7.99 ± 0.03 f	79.6 ± 0.27 a	78.58 ± 0.27 a	13.61 ± 0.28 f	12.61 ± 0.06 h	5.14 ± 0.11 cd	5.25 ± 0.02 e
DI×MLE^−^×Nano-Zn100	8.86 ± 0.05 f	8.66 ± 0.05 e	79.9 ± 0.19 a	78.88 ± 0.19 a	14.10 ± 0.54 ef	13.66 ± 0.12 g	5.32 ± 0.20 bc	5.69 ± 0.05 d
DI×MLE^+^×Nano-Zn0	8.01 ± 0.03 g	7.81 ± 0.03 f	75.9 ± 0.60 a	74.90 ± 0.60 a	12.31 ± 0.49 g	11.70 ± 0.13 i	4.65 ± 0.19 e	4.87 ± 0.05 f
DI×MLE^+^×Nano-Zn50	9.91 ± 0.19 d	9.37 ± 0.25 d	80.0 ± 0.29 a	79.01 ± 0.29 a	14.82 ± 0.13 df	15.34 ± 0.29 e	5.59 ± 0.05 ab	6.39 ± 0.12 b
DI×MLE^+^×Nano-Zn100	10.46 ± 0.06 c	10.11 ± 0.04 c	80.6 ± 0.12 a	79.93 ± 0.25 a	15.37 ± 0.31 d	16.27 ± 0.05 d	5.80 ± 0.12 a	6.78 ± 0.02 a

^#^ Values are means of (n = 5) ± standard error. Mean values in each column followed by a different lowercase letter are significantly different by Duncan’s multiple range test at *p* ≤ 0.05.

**Table 3 plants-14-00544-t003:** Oxidative stress indicator (H_2_O_2_), tissue relative water content (RWC), and membrane stability index (MSI) responses to exogenously applied ZnO nanoparticles (Nano-Zn) and moringa leaf extract (MLE) under normal (FI) and drought (DI) conditions during the 2021 and 2022 seasons.

Source of Variation	H_2_O_2_ (µmol g^−1^ FW)		MSI (%)		RWC (%)	
	2021	2022	2021	2022	2021	2022
**Irrigation**						
FI	4.50 ± 0.14 b ^#^	4.60 ± 0.14 b	77.3 ± 1.17 a	75.3 ± 1.18 a	73.1 ± 1.21 a	71.2 ± 1.25 a
DI	6.50 ± 0.19 a	6.60 ± 0.21 a	69.4 ± 1.69 b	67.1 ± 1.81 b	62.5 ± 1.38 b	60.9 ± 1.50 b
**MLE**						
MLE^−^	5.10 ± 0.25 b	5.30 ± 0.25 b	70.5 ± 1.96 b	68.3 ± 2.00 b	64.4 ± 1.61 b	62.6 ± 1.71 b
MLE^+^	5.90 ± 0.31 a	6.00 ± 0.33 a	76.3 ± 1.54 a	74.2 ± 1.54 b	71.1 ± 1.27 a	69.5 ± 1.36 a
**Nano-Zn**						
Nano-Zn0	4.70 ± 0.29 c	4.80 ± 0.28 c	67.3 ± 2.28 c	64.6 ± 2.36 c	61.2 ± 1.60 c	59.1 ± 1.74 c
Nano-Zn25	5.70 ± 0.32 b	5.80 ± 0.34 b	75.6 ± 1.65 b	73.8 ± 1.62 b	69.7 ± 1.32 b	68.0 ± 1.32 b
Nano-Zn50	6.10 ± 0.36 a	6.30 ± 0.38 a	77.2 ± 1.83 a	75.4 ± 1.64 a	72.4 ± 1.59 a	71.1 ± 1.59 a
**Irrigation×MLE×Nano-Zn**						
FI×MLE^−^×Nano-Zn0	5.35 ± 0.04 g	5.36 ± 0.04 g	68.4 ± 0.31 fg	65.9 ± 0.18 g	65.0 ± 0.54 g	62.7 ± 0.16 g
FI×MLE^−^×Nano-Zn50	4.37 ± 0.04 j	4.89 ± 0.07 i	77.7 ± 0.32 d	75.8 ± 0.32 d	72.2 ± 0.30 d	70.2 ± 0.30 d
FI×MLE^−^×Nano-Zn100	4.68 ± 0.05 i	4.40 ± 0.06 j	79.2 ± 0.56 c	77.3 ± 0.35 c	74.5 ± 0.43 c	72.9 ± 0.43 c
FI×MLE^+^×Nano-Zn0	5.03 ± 0.06 h	5.09 ± 0.03 h	74.0 ± 0.20 e	72.2 ± 0.18 e	69.9 ± 0.47 e	68.6 ± 0.47 e
FI×MLE^+^×Nano-Zn50	3.99 ± 0.01 k	4.08 ± 0.02 k	81.1 ± 0.08 b	79.3 ± 0.08 b	76.8 ± 0.27 b	74.8 ± 0.27 b
FI×MLE^+^×Nano-Zn100	3.61 ± 0.01 l	3.73 ± 0.00 l	83.4 ± 0.42 a	81.5 ± 0.42 a	79.9 ± 0.90 a	77.8 ± 0.90 a
DI×MLE^−^×Nano-Zn0	7.78 ± 0.02 a	8.08 ± 0.02 a	59.3 ± 0.76 h	56.1 ± 0.58 i	49.6 ± 0.37 j	47.2 ± 0.52 j
DI×MLE^−^×Nano-Zn50	6.54 ± 0.03 c	6.69 ± 0.04 c	69.2 ± 0.38 f	67.4 ± 0.38 f	62.1 ± 0.34 h	60.1 ± 0.34 h
DI×MLE^−^×Nano-Zn100	6.09 ± 0.03 d	6.21 ± 0.05 d	69.0 ± 0.30 f	67.2f ± 0.28 g	63.1 ± 0.38 h	62.8 ± 0.38 g
DI×MLE^+^×Nano-Zn0	7.18 ± 0.01 b	7.31 ± 0.01 b	67.4 ± 0.26 g	64.3 ± 0.21 h	60.3 ± 0.54 i	57.9 ± 0.44 i
DI×MLE^+^×Nano-Zn50	5.78 ± 0.04 e	5.81 ± 0.04 e	74.4 ± 0.47 e	72.6 ± 0.20 e	67.8 ± 0.13 f	66.8 ± 0.13 f
DI×MLE^+^×Nano-Zn100	5.57 ± 0.01 f	5.64 ± 0.09 f	77.2 ± 0.23 d	75.4 ± 0.23 d	71.9 ± 0.11 d	70.9 ± 0.11 d

^#^ Values are means of (n = 5) ± standard error. Mean values in each column followed by a different lowercase letter are significantly different by Duncan’s multiple range test at *p* ≤ 0.05.

**Table 4 plants-14-00544-t004:** Photosynthetic pigment (chlorophyll a and b, carotenoids) content responses to exogenously applied ZnO nanoparticles (Nano-Zn) and moringa leaf extract (MLE) under normal (FI) and drought (DI) conditions during the 2021 and 2022 seasons.

Source of Variation	Chlorophyll a (mg g^−1^ FW)		Chlorophyll b (mg g^−1^ FW)		Carotenoids (mg g^−1^ FW)	
	2021	2022	2021	2022	2021	2022
**Irrigation**						
FI	2.38 ± 0.05 a ^#^	2.24 ± 0.04 a	1.61 ± 0.06 a	1.49 ± 0.05 a	0.75 ± 0.02 a	0.68 ± 0.02 a
DI	1.80 ± 0.07 b	1.68 ± 0.08 b	1.05 ± 0.07 b	0.94 ± 0.07 b	0.65 ± 0.04 b	0.41 ± 0.03 b
**MLE**						
MLE^−^	1.97 ± 0.09 b	1.83 ± 0.09 b	1.19 ± 0.09 b	1.09 ± 0.08 b	0.63 ± 0.04 b	0.49 ± 0.04 b
MLE^+^	2.22 ± 0.09 a	2.09 ± 0.09 a	1.47 ± 0.09 a	1.34 ± 0.08 a	0.78 ± 0.04 a	0.60 ± 0.04 a
**Nano-Zn**						
Nano-Zn0	1.82 ± 0.10 c	1.68 ± 0.11 c	1.02 ± 0.10 c	0.94 ± 0.10 c	0.55 ± 0.04 c	0.43 ± 0.05 c
Nano-Zn25	2.16 ± 0.10 b	2.03 ± 0.10 b	1.41 ± 0.10 b	1.29 ± 0.09 b	0.75 ± 0.02 b	0.57 ± 0.04 b
Nano-Zn50	2.30 ± 0.09 a	2.17 ± 0.08 a	1.55 ± 0.08 a	1.42 ± 0.08 a	0.81 ± 0.02 a	0.63 ± 0.04 a
**Irrigation×MLE×Nano-Zn**						
FI×MLE^−^×Nano-Zn0	2.07 ± 0.02 f	1.96 ± 0.01 f	1.21 ± 0.01 g	1.16 ± 0.02 g	0.607 ± 0.01 f	0.274 ± 0.01 j
FI×MLE^−^×Nano-Zn50	2.33 ± 0.01 d	2.20 ± 0.01 d	1.57 ± 0.01 d	1.45 ± 0.01 d	0.727 ± 0.00 d	0.345 ± 0.01 i
FI×MLE^−^×Nano-Zn100	2.42 ± 0.01 c	2.28 ± 0.01 c	1.68 ± 0.01 c	1.53 ± 0.01 c	0.780 ± 0.01 c	0.432 ± 0.01 h
FI×MLE^+^×Nano-Zn0	2.23 ± 0.01 e	2.10 ± 0.01 e	1.47 ± 0.01 e	1.35 ± 0.01 e	0.675 ± 0.00 e	0.273 ± 0.01 j
FI×MLE^+^×Nano-Zn50	2.54 ± 0.01 b	2.40 ± 0.01 b	1.80 ± 0.01 b	1.65 ± 0.01 b	0.838 ± 0.00 b	0.542 ± 0.00 g
FI×MLE^+^×Nano-Zn100	2.66 ± 0.01 a	2.51 ± 0.01 a	1.92 ± 0.01 a	1.76 ± 0.01 a	0.902 ± 0.00 a	0.613 ± 0.00 e
DI×MLE^−^×Nano-Zn0	1.41 ± 0.01 j	1.23 ± 0.01 j	0.61 ± 0.01 k	0.55 ± 0.02 k	0.305 ± 0.01 g	0.582 ± 0.01 f
DI×MLE^−^×Nano-Zn50	1.68 ± 0.02 h	1.55 ± 0.02 h	0.93 ± 0.02 i	0.83 ± 0.01 i	0.667 ± 0.00 e	0.647 ± 0.01 d
DI×MLE^−^×Nano-Zn100	1.88 ± 0.02 g	1.75 ± 0.02 g	1.13 ± 0.02 h	1.00 ± 0.02 h	0.720 ± 0.01 d	0.685 ± 0.01 c
DI×MLE^+^×Nano-Zn0	1.55 ± 0.02 i	1.42 ± 0.02 i	0.80 ± 0.02 j	0.69 ± 0.01 j	0.615 ± 0.00 f	0.597 ± 0.01 ef
DI×MLE^+^×Nano-Zn50	2.08 ± 0.01 f	1.97 ± 0.01 f	1.33 ± 0.01 f	1.22 ± 0.01 f	0.778 ± 0.00 c	0.745 ± 0.00 b
DI×MLE^+^×Nano-Zn100	2.23 ± 0.02 e	2.14 ± 0.01 e	1.48 ± 0.02 e	1.37 ± 0.01 e	0.842 ± 0.00 b	0.802 ± 0.01 a

^#^ Values are means of (n = 5) ± standard error. Mean values in each column followed by a different lowercase letter are significantly different by Duncan’s multiple range test at *p* ≤ 0.05.

**Table 5 plants-14-00544-t005:** Photosynthetic efficiency responses to exogenously applied ZnO nanoparticles (Nano-Zn) and moringa leaf extract (MLE) under normal (FI) and drought (DI) conditions during the 2021 and 2022 seasons.

Source of Variation	*Fv*/*Fm*		PI	
	2021	2022	2021	2022
**Irrigation**				
FI	0.817 ± 0.00 a ^#^	0.809 ± 0.00 a	5.76 ± 0.11 a	6.71 ± 0.12 a
DI	0.774 ± 0.01 b	0.766 ± 0.01 b	4.82 ± 0.13 b	5.63 ± 0.17 b
**MLE**				
MLE^−^	0.782 ± 0.01 b	0.772 ± 0.01 b	4.90 ± 0.15 b	5.74 ± 0.19 b
MLE^+^	0.809 ± 0.01 a	0.803 ± 0.01 a	5.67 ± 0.13 a	6.60 ± 0.13 a
**Nano-Zn**				
Nano-Zn0	0.769 ± 0.01 c	0.756 ± 0.01 c	4.94 ± 0.21 c	5.70 ± 0.28 c
Nano-Zn25	0.802 ± 0.01 b	0.795 ± 0.01 b	5.28 ± 0.18 b	6.28 ± 0.18 b
Nano-Zn50	0.817 ± 0.01 a	0.811 ± 0.01 a	5.63 ± 0.18 a	6.53 ± 0.18 a
**Irrigation×MLE×Nano-Zn**				
FI×MLE^−^×Nano-Zn0	0.791 ± 0.00 g	0.786 ± 0.00 e	5.14 ± 0.06 e	5.84 ± 0.06 g
FI×MLE^−^×Nano-Zn50	0.804 ± 0.00 e	0.795 ± 0.00 d	5.44 ± 0.02 d	6.44 ± 0.02 d
FI×MLE^−^×Nano-Zn100	0.822 ± 0.00 c	0.813 ± 0.00 c	5.67 ± 0.02 c	6.67 ± 0.02 c
FI×MLE^+^×Nano-Zn0	0.806 ± 0.00 e	0.792 ± 0.00 d	5.76 ± 0.04 c	6.76 ± 0.04 c
FI×MLE^+^×Nano-Zn50	0.836 ± 0.00 b	0.829 ± 0.00 b	6.10 ± 0.04 b	7.43 ± 0.02 a
FI×MLE^+^×Nano-Zn100	0.845 ± 0.00 a	0.840 ± 0.00 a	6.43 ± 0.18 a	7.10 ± 0.03 b
DI×MLE^−^×Nano-Zn0	0.720 ± 0.00 j	0.698 ± 0.00 i	3.87 ± 0.05h	4.19 ± 0.01 i
DI×MLE^−^×Nano-Zn50	0.772 ± 0.00 h	0.763 ± 0.00 g	4.49 ± 0.11 g	5.49 ± 0.05 h
DI×MLE^−^×Nano-Zn100	0.785 ± 0.00 g	0.778 ± 0.00 f	4.82 ± 0.07 f	5.82 ± 0.07 g
DI×MLE^+^×Nano-Zn0	0.759 ± 0.00 i	0.749 ± 0.00 h	4.99 ± 0.03 ef	5.99 ± 0.03 f
DI×MLE^+^×Nano-Zn50	0.797 ± 0.00 f	0.793 ± 0.00 d	5.12 ± 0.03 e	6.10 ± 0.02 e
DI×MLE^+^×Nano-Zn100	0.813 ± 0.00 d	0.814 ± 0.00 c	5.61 ± 0.06 cd	6.21 ± 0.00 e

^#^ Values are means of (n = 5) ± standard error. Mean values in each column followed by a different lowercase letter are significantly different by Duncan’s multiple range test at *p* ≤ 0.05.

**Table 6 plants-14-00544-t006:** Osmoprotectant responses to exogenously applied ZnO nanoparticles (Nano-Zn) and moringa leaf extract (MLE) under normal (FI) and drought (DI) conditions during the 2021 and 2022 seasons.

Source of Variation	Total Soluble Sugars (mg g^−1^ DW)		Free Amino Acids (mg g^−1^ DW)	
	2021	2022	2021	2022
**Irrigation**				
FI	17.3 ± 0.54 b ^#^	17.6 ± 0.54 b	9.40 ± 0.32 b	9.60 ± 0.32 b
DI	28.1 ± 1.12 a	28.3 ± 1.09 a	12.9 ± 0.19 a	13.1 ± 0.20 a
**MLE**				
MLE^−^	20.9 ± 1.22 b	21.2 ± 1.24 b	10.6 ± 0.49 b	10.8 ± 0.48 b
MLE+	24.5 ± 1.77 a	24.8 ± 1.72 a	11.7 ± 0.47 a	11.9 ± 0.48 a
**Nano-Zn**				
Nano-Zn0	19.2 ± 1.44 c	19.5 ± 1.44 c	10.0 ± 0.64 c	10.2 ± 0.64 c
Nano-Zn25	23.1 ± 1.64 b	23.4 ± 1.67 b	11.4 ± 0.54 b	11.5 ± 0.53 b
Nano-Zn50	25.8 ± 2.21 a	25.9 ± 2.14 a	12.1 ± 0.50 a	12.3 ± 0.52 a
**Irrigation×MLE×Nano-Zn**				
FI×MLE^−^×Nano-Zn0	13.5 ± 0.06 l	13.7 ± 0.22 i	7.37 ± 0.06 k	7.59 ± 0.21 l
FI×MLE^−^×Nano-Zn50	17.1 ± 0.07 j	17.4 ± 0.09 g	9.19 ± 0.30 i	9.33 ± 0.15 j
FI×MLE^−^×Nano-Zn100	18.2 ± 0.09 i	18.5 ± 0.09 g	9.79 ± 0.03 h	9.87 ± 0.02 i
FI×MLE+×Nano-Zn0	15.6 ± 0.08 k	16.0 ± 0.12 h	8.51 ± 0.05 j	8.61 ± 0.04 k
FI×MLE+×Nano-Zn50	19.3 ± 0.09 h	19.6 ± 0.11 f	10.2 ± 0.07 g	10.4 ± 0.08 h
FI×MLE+×Nano-Zn100	20.1 ± 0.06 g	20.3 ± 0.05 f	11.9 ± 0.05 f	11.7 ± 0.03 g
DI×MLE^−^×Nano-Zn0	23.5 ± 0.06 f	23.6 ± 0.04 e	11.9 ± 0.04 e	12.1 ± 0.05 f
DI×MLE^−^×Nano-Zn50	24.8 ± 0.03 d	24.9 ± 0.05 d	12.6 ± 0.05 cd	12.7 ± 0.03 d
DI×MLE^−^×Nano-Zn100	28.2 ± 0.09 c	28.9 ± 0.35 c	12.9 ± 0.04 c	13.1 ± 0.06 c
DI×MLE+×Nano-Zn0	24.4 ± 0.06 e	24.8 ± 0.31 d	12.3 ± 0.06 d	12.4 ± 0.03 e
DI×MLE+×Nano-Zn50	31.2 ± 0.12 b	31.8 ± 0.27 b	13.6 ± 0.05 b	13.7 ± 0.04 b
DI×MLE+×Nano-Zn100	36.7 ± 0.09 a	36.0 ± 1.01 a	14.2 ± 0.13 a	14.5 ± 0.08 a

^#^ Values are means of (n = 5) ± standard error. Mean values in each column followed by a different lowercase letter are significantly different by Duncan’s multiple range test at *p* ≤ 0.05.

**Table 7 plants-14-00544-t007:** The soil physicochemical analysis in the experimental site.

Soil Property	Value
Sand (%)	72.2
Silt (%)	14.4
Clay (%)	13.4
Texture class	SL
ρd (g.cm^−3^)	1.76
K_sat_ (cm h^−1^)	2.86
CEC (cmol kg^−1^)	11.67
WP (%)	10.83
AW (%)	12.50
FC (%)	24.21
pH	7.68
ECe (dS m^−1^)	6.45
CaCO_3_ (%)	8.67
OM %	0.92

SL: sandy loam; ρd: bulk density; K_sat_; hydraulic conductivity; CEC: cation exchange capacity; WP, wilting point; AW, available water; FC, field capacity; OM: organic matter ECe; electrical conductivity.

**Table 8 plants-14-00544-t008:** Chemical characterization of *Moringa oleifera* leaf extract (on a dry weight basis) components.

Component	Unit	Value
**Osmoprotectants**		
Total amino acids	g g^−1^ DW	0.16
Proline	g g^−1^ DW	0.02
Total soluble sugars		0.22
**Mineral nutrients**		
Nitrogen (N)	g kg^−1^ DW	30.9
Magnesium (Mg)	g kg^−1^ DW	5.85
Potassium (K)	g kg^−1^ DW	23.2
Calcium (Ca)	g kg^−1^ DW	7.71
Phosphorus (P)	g kg^−1^ DW	4.19
Sulphur (S)	g kg^−1^ DW	2.33
Manganese (Mn)	g kg^−1^ DW	0.87
Copper (Cu)	g kg^−1^ DW	0.23
Zinc (Zn)	g kg^−1^ DW	0.49
Iron (Fe)	g kg^−1^ DW	1.09
**Antioxidants**		
Salicylic acid	µg g^−1^ DW	92.4
Tocopherol	µg g^−1^ DW	21.5
Glutathione (GSH)	µmol GSH g^−1^ DW	0.36
Ascorbic acid (AsA; Vitamin C)	µmol AsA g^−1^ DW	2.18
DPPH radical-scavenging activity	%	76.3
**Phytohormones**		
Auxins	µg g^−1^ DW	2.06
Gibberellins	µg g^−1^ DW	1.99
Cytokinins	µg g^−1^ DW	2.39

## Data Availability

The original contributions presented in this study are included in the article.

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
