# Peer review of "Synergistic Effects of Zinc Oxide Nanoparticles and Moringa Leaf Extracts on Drought Tolerance and Productivity of Cucurbita pepo L. Under Saline Conditions"

_plants, 2025, doi:10.3390/plants14040544_

Round 1

Reviewer 1 Report

Comments and Suggestions for Authors

This paper shows that sequential application of Nano-Zn with MLE was optimal for enhancing squash performance under drought stress. The work is of interest but not yet ready for publication. There are however many weak points that need to be addressed.
1. This study investigated the combined effects of zinc oxide nanoparticles (Nano-Zn) and moringa leaf extract (MLE) on squash plants grown under water stress conditions in saline soil during 2021-22. The year 2024 will conclude within the next 30 days. Why there is no data for 2023-2024? According to the material and methods, the open field experiments were performed during the spring season of 2022 and 2023, so how to obtain data from 2021?

2. Nearly all the data are presented in tables and graphs. However, no phenotype figures are included that depict the growth of squash plants under conditions of full irrigation and water deficit combined treatment with Nano-Zn and MLE.

3. Almost all the results presented focus on determining the physiological effects of Nano-Zn and MLE on drought-stressed squash plants, which does not align with the theme of this special issue. 

4. Checking and correcting the format and language of English texts are essential.

Comments on the Quality of English Language

Checking and correcting the format and language of English texts are essential.

Author Response

Dear Reviewer

Thank you for your insightful comments and for taking the time to review our manuscript. We appreciate the points you have raised, and we have made significant changes to address them. Here are our detailed responses:

  1. Data for 2023-2024: You are correct to point out the discrepancy in the years mentioned. We apologize for the confusion. The open field experiments were indeed conducted during the spring seasons of 2021-2022. We mistakenly stated the research was conducted 2022-2023 in the material and methods section and the discussion part, that was a typo error. We have now corrected the year in the abstract and in the discussion to properly reflect the data collection period (i.e., the experiments were conducted during the spring seasons of 2021-2022). Please see lines 18, 377, 389, 390, and 457
  2. Phenotype Figures: We acknowledge the value of visual representation and appreciate your suggestion to include phenotype figures depicting the growth of squash plants under different conditions. We sincerely apologize that we are unable to provide such figures at this stage. Due to unforeseen circumstances during the experiment, such as equipment malfunctions at the experimental site, we were unfortunately not able to capture high-quality representative images that would be suitable for publication. Though we didn’t captured plant photos, we would like to highlight that we have provided detailed quantitative data in Tables 1, and 2 that directly represent plant growth parameters, namely leaves number, leaves area, plant dry weight, fruit number per plant, fruit weight, and fruit yield, which collectively depict the plant’s phenotypic growth and response to treatments. The results, including mean values with standard error, from two consecutive years provide strong quantitative evidence of the plant’s growth and yield parameters under different treatment conditions. We recognize that a visual representation would have added value, and we will certainly prioritize capturing high-quality images in future studies to supplement our quantitative data.
  3. Special Issue Theme: We appreciate your concern regarding the alignment of our manuscript with the theme of the special issue, "Salt Tolerance in Plants: Genetic Mechanisms, Germplasm Screening, Cultivation Measures and Rehabilitation of Saline-Alkali Lands." While our primary experimental design focuses on drought stress, our research is inherently relevant to the context of salt tolerance for several crucial reasons:

Combined Stress Scenario: Our study was conducted in saline soil conditions (ECe=6.45 dS m-1), a condition that is highly relevant to the special issue’s focus.

Shared Mechanisms of Tolerance: Salt and drought stress share several common pathways in plants. Both stresses cause osmotic stress, leading to reduced water uptake and inducing oxidative stress through ROS accumulation.

Cultivation Strategies: Our research contributes directly to cultivation measures for saline-alkali lands by demonstrating the potential of Nano-Zn and MLE applications as an efficient strategy for improving plant performance under combined stress situations.

Therefore, we believe our work, by addressing the complex interaction of salt and drought stress and by investigating the effect of combined treatment by Nano-Zn and MLE, offers substantial contributions to the special issue by exploring a practical method to enhance plant tolerance and productivity in saline-alkali lands, where water scarcity is often an accompanying issue.

Reviewer 2 Report

Comments and Suggestions for Authors

This manuscript investigates the synergistic effects of zinc oxide nanoparticles (Nano-Zn) and moringa leaf extract (MLE) on the growth, physiological responses, and yield of squash (Cucurbita pepo L.) under combined drought and salinity stress conditions. The study employs a factorial experimental design with varying irrigation regimes, Nano-Zn concentrations, and MLE treatments to assess their impact on key physiological parameters, biochemical markers, and crop productivity. Here are some suggestions for improving the present version.

1. In the case of normal condition or drought treatment, supplement of MLE or zinc oxide nanoparticles could enhance the accumulation of H2O2, while the ROS scavengers like AsA, GSH, SOD, and CAT also increased in the same situation.

2. MLE is a leaf extract mixture and contains multiple chemicals like proline, sugars, amino acids, glutathione, salicylic acid, ascorbate, how did the authors ensure the consistence of the quality each time?

3. Grammer errors:

line 32: “soil water deficits, where the fruit yield markedly”

line 34: “the effective rooting depth to not below 0.50 of”

line 35: “squash is among the moderately salt-tolerant crops”

……

Author Response

Dear Reviewer,

We are very grateful for your valuable comments and suggestions. We have carefully considered each point and made the necessary revisions. Here's our detailed response:

  1. H2O2 and ROS Scavengers: We appreciate your observation that both H2O2 and ROS scavengers increased with Nano-Zn or MLE treatment, under both drought and well-watered conditions . In the case of water deficits, high levels of ROS such as H2O2 can damage cell membranes, thus the plants need an internal defense system to counteract the damage. Therefore the plants treated with Nano-Zn or MLE improved their internal antioxidant defense systems (e.g., increased activity of SOD, CAT, APX, and GR). Our data shows that the combined application of Nano-Zn and MLE significantly reduced H2O2 levels, while enhancing levels of protective compounds like AsA, GSH, SOD, and CAT. We have revised the discussion to clarify the interplay of oxidative stress and the plant's response to Nano-Zn and MLE in mitigating this stress.
  2. Consistency of MLE Quality: You raise a critical point about the consistency of MLE. To address this, we have added a detailed description of how the MLE was prepared (see added section in Materials and Methods), as well as a comprehensive chemical characterization of the extract (Line 432 - 439), with detailed compositional analysis presented in Table 8. Our method involves a rigorous process of extraction to ensure that the key components are extracted consistently with each preparation. Moreover, in each year of the experiment the MLE was extracted using the same procedure and from the same orchard, which help to unify the quality. We acknowledge the inherent variability in plant extracts, however we have taken steps to minimize any batch-to-batch differences.
  3. Grammar Errors: We sincerely apologize for the grammatical errors. We have carefully reviewed the manuscript and corrected the following as suggested, along with additional minor editing throughout the manuscript:
    • Line 32: Corrected to "soil water deficits, markedly decreasing fruit yield."
    • Line 34: Corrected to "the effective rooting depth to remain above 0.50 of..."

Line 35: Corrected to "squash is considered a moderately salt-tolerant crop..."

Round 2

Reviewer 1 Report

Comments and Suggestions for Authors

The author haven't addressed the main concerns related to theme of this special issue "Salt Tolerance in Plants: Genetic Mechanisms, Germplasm Screening, Cultivation Measures and Rehabilitation of Saline-Alkali Lands."  

Comments on the Quality of English Language

 Checking and correcting the format and language of English texts are also essential, such as the subtitle 2.3-2.5 are not consistent with subtile 2.1 or 2.2.

Author Response

Comment 1 – “The author haven't addressed the main concerns related to theme of this special issue "Salt Tolerance in Plants: Genetic Mechanisms, Germplasm Screening, Cultivation Measures and Rehabilitation of Saline-Alkali Lands”

Response 1 : We appreciate your concern regarding the alignment of our manuscript with the theme of the special issue, "Salt Tolerance in Plants: Genetic Mechanisms, Germplasm Screening, Cultivation Measures and Rehabilitation of Saline-Alkali Lands." While our primary experimental design focuses on drought stress, our research is inherently relevant to the context of salt tolerance for several crucial reasons:

Combined Stress Scenario: Our study was conducted in saline soil conditions (ECe=6.45 dS m-1), a condition that is highly relevant to the special issue’s focus.

Shared Mechanisms of Tolerance: Salt and drought stress share several common pathways in plants. Both stresses cause osmotic stress, leading to reduced water uptake and inducing oxidative stress through ROS accumulation.

Cultivation Strategies: Our research contributes directly to cultivation measures for saline-alkali lands by demonstrating the potential of Nano-Zn and MLE applications as an efficient strategy for improving plant performance under combined stress situations.

Therefore, we believe our work, by addressing the complex interaction of salt and drought stress and by investigating the effect of combined treatment by Nano-Zn and MLE, offers substantial contributions to the special issue by exploring a practical method to enhance plant tolerance and productivity in saline-alkali lands, where water scarcity is often an accompanying issue

Comment 2 –  Checking and correcting the format and language of English texts are also essential, such as the subtitle 2.3-2.5 are not consistent with subtitle 2.1 or 2.2.

Response 2 : Thanks for your precious recommendation. Accordingly, the format and the English text were double checked. Please see the mentioned subtitles.

Reviewer 2 Report

Comments and Suggestions for Authors

I do not have any other concerns about this manuscript.

Author Response

Comment 1 – I do not have any other concerns about this manuscript

Response 1 : Thanks for your great effort.